# Effect of Fermented Soymilk-Honey from Different Probiotics on Osteocalcin Level in Menopausal Women

**DOI:** 10.3390/nu13103581

**Published:** 2021-10-13

**Authors:** Sri Desfita, Wulan Sari, Yusmarini Yusmarini, Usman Pato, Małgorzata Zakłos-Szyda, Grażyna Budryn

**Affiliations:** 1Public Health Program, STIKes Hang Tuah Pekanbaru, Pekanbaru 28282, Riau, Indonesia; wulan.sari71@gmail.com; 2Faculty of Agriculture, Universitas Riau, Pekanbaru 28293, Riau, Indonesia; marini_thp@yahoo.co.id (Y.Y.); usmanpato@yahoo.com (U.P.); 3Institute of Molecular and Industrial Biotechnology, Faculty of Biotechnology and Food Sciences, Lodz University of Technology, Stefanowskiego 4/10, 90-924 Lodz, Poland; malgorzata.zaklos-szyda@p.lodz.pl; 4Institute of Food Technology and Analysis, Faculty of Biotechnology and Food Sciences, Lodz University of Technology, Stefanowskiego 4/10, 90-924 Lodz, Poland; grazyna.budryn@p.lodz.pl

**Keywords:** osteocalcin, honey, *Lactobacillus casei*, *Lactobacillus plantarum*, fermented soymilk, menopause

## Abstract

Osteoporosis has been discovered to be a risk factor for menopausal women. Although synbiotics (probiotics and prebiotics) are found in fermented soymilk-honey made using local probiotics, their effect on osteocalcin levels is still unknown. Therefore, this study’s objective was to determine the influence of fermented soymilk-honey from different probiotics on osteocalcin levels. A 90-day pre–post quasi-experimental study with a control design was conducted on 54 postmenopausal women divided into three intervention groups namely, the soymilk (SM) group, the soymilk-honey fermented with *Lactobacillus casei* subsp. *casei* R-68 (SMH Lc) group, and the soymilk-honey fermented with *Lactobacillus plantarum* 1 R 1.3.2 (SMH Lp) group. Participants consumed 100 mL of soymilk (SM) or fermented soymilk with honey (SMH Lc or SMH Lp) for 90 days. At the beginning and end of the study, the blood serum osteocalcin level was measured and subjects’ health status was assessed, such as cholesterol total, random blood glucose, and uric acid levels. Our results presented that in the SMH Lp group, 90 days supplementation of soy-honey milk fermented with *Lactobacillus plantarum* 1 R 1.3.2 significantly reduced the level of blood serum osteocalcin. Based on these results it is justified to perform more detailed studies on the effect of fermented soy-honey milk on bone health.

## 1. Introduction

Osteoporosis is a chronic disease characterized by a decrease in the density and strength of the bones, which rises risk of fractures. This disease is associated with increased morbidity, mortality, and higher health care costs [1]. Every year more than 8.9 million fractures occur due to osteoporosis, with an estimated effect on 200 million women globally [2]. Menopausal women are prone to osteoporosis as a result of a decreased estrogen hormone and reduced bone mass (an average of 2–5% per year) [3]. Osteoporosis is a public health problem and requires effective prevention methods [1,4]. Therefore, osteoporosis patients with low calcium intake or absorption, vitamin D deficiency, or on pharmaceutical treatment should take vitamin D and calcium supplements. However, calcium and vitamin D supplementation can cause some side effects such as digestive tract disorders and kidney stones, hence its effectiveness is still debatable [5].

Despite the widespread awareness of the importance of calcium intake in osteoporosis prevention, milk consumption as the primary dietary source and supplement has declined in recent years [6]. In addition, due to allergies, lactose intolerance, or vegetarianism, specific populations cannot consume dairy products [7]. Consequently, there is growing need to development of dairy alternatives derived from plants to improve bone health with minimal side effects.

Till now, several studies have been performed on the role of probiotics and prebiotics in reducing the effects of menopause. Probiotics are defined as living microorganisms that are beneficial to the health of the host once administered in adequate quantities. Meanwhile, prebiotics are food components that occur naturally in plants or as enzymatically produced synthetic polysaccharides, and are included in the dietary fiber. These components can benefit the health of the host by stimulation of the growth of health promoting microorganisms. In individuals with low calcium consumption, prebiotics are an effective way to enhance calcium absorption and improve bone mineral density. Thus, the combination of probiotics and prebiotics, called synbiotics, can provide a synergistic effect that is beneficial to well-being [8,9].

Soybeans and honey have been recognized as functional foods due to the presence of bioactive components. Currently, several studies have developed soy milk as a medium for probiotics and prebiotics not only to improve health benefits, but also taste. Oligosaccharides present in soybeans, especially stachyose, are able to stimulate the growth of bifidobacteria, therefore they are characterized as prebiotics [10]. Furthermore, the high content of isoflavones resembling the estrogen structure has the potential of reducing the risk of osteoporosis, menopausal symptoms, and breast cancer [7,11]. What is more, the nutritional value and concentration of bioactive compounds in soybeans can be increased through fermentation. Soybean fermentation with β-glucosidase producing bacteria can convert glucoside isoflavones into aglycone isoflavones, which possess higher biological activity and bioavailability [12,13]. On the other hand, honey is a functional food with unique composition that contains fructooligosaccharides acting as prebiotics. Therefore, the addition of honey to fermented milk can increase the growth of bifidobacteria and extend the shelflife of the product via the enhanced lactic acid production and pH lowering [14]. *Lactobacillus acidophilus* counts were improved in formulas containing honey, soy milk, and aloevera [15]. The total of *Lactobacillus acidophilus* also increased in Rayeb’s milk (consisting of 50% cow’s milk and 50% soy milk) after supplementing with honey [16]. Since probiotics and prebiotics (together known as synbiotics) are important diet components, together with fermented soy milk and honey can reveal a strong influence on the bone health improvement in postmenopausal women.

International Osteoporosis Foundation and European Calcified Tissue Society Working Group have recommended serum procollagen type I N-terminal propeptide (PINP) and collagen type I C-terminal telopeptide (CTX) as bone turnover markers [17]. However, the choice of bone marker for monitoring therapy may depend on the type of antiresorptive agent used and in some situations osteocalcin can be used. Osteocalcin comprises of noncollagenous protein with 49 amino acids produced by osteoblasts, odontoblasts, and hypertrophic chondrocytes [18]. This is one of the essential biochemical indicators for bone formation [19]. In the bone mineralization process, osteocalcin is bound to hydroxyapatite in the bone matrix, and only a limited part in the circulation is released [18]. Osteoporosis decreases the formation of hydroxyapatite crystals, and as a result, increases serum osteocalcin levels [20]. Moreover, osteocalcin level is increased in bone diseases such as osteoporosis, acromegaly, and bone metastases [21]. Osteopenic and osteoporotic women had higher serum osteocalcin than women with normal bone mass density [20]. An observational study in China showed that the risk of osteoporosis was significantly related to the age, age of menopause, duration of menopause, BMI, and educational level of postmenopausal women. After adjusting for age, height, weight, and duration of menopause, the data showed that high levels of osteocalcin were associated with lower bone mass density [22].

Several animal studies have reported the role of probiotics and prebiotics on bone health [23,24,25], also probiotics have been shown to benefit bone mass sustaining in women [4,26,27]. However, the majority of these studies used probiotics as supplements and dairy foods without the addition of honey. For this reason, this study aimed to examine the effect of fermented soy-honey milk using different probiotics (*Lactobacillus casei* subsp. *casei* R-68 and *Lactobacillus plantarum* 1 R. 1.3.2) on blood serum osteocalcin levels in postmenopausal women.

## 2. Materials and Methods

### 2.1. Study Design

This was a quasi-experimental, pre–post study with a control design to determine the effect of fermented soy-honey milk on osteocalcin levels in postmenopausal women. A quasi-experiment was used because of difficulty identifying the participants that liked the three types of treatment, and hence randomizing the participants was impractical. Consequently, based on their preferences, the participants were allotted into the sort of intervention. Therefore, the participants were able to maintain their participation during the 90-day intervention period.

The blood serum osteocalcin standard deviation in both groups was 8 ng/mL [4] with a significance level of 5% (Z_1−α_ = 1.96) and a test power of 90% (Z_1−β_ = 1.28), while the expected clinical difference between the two groups was 10. At least 13 samples were obtained for each group, and 50% of the minimum sample size was added to anticipate the participants that drop out during the study. Therefore, each group received 20 samples for a total of 60 research subjects.

### 2.2. Preparation of Synbiotics (Soymilk, Honey, and Lactic Acid Bacteria)

Soymilk, *Lactobacillus casei* subsp. casei R-68, *Lactobacillus plantarum* 1 R 1.3.2, and honey were used as intervention materials. Bacteria were selected from collection of Agricultural Products Processing Laboratory of Riau University’s Agricultural College and were obtained through the process of bacterial propagation. The purified honey from Rupat Island (Bengkalis Regency, Riau Province) that had a low water content (19.1%) and a high calcium content (270 mg/kg) was used. The soy-honey milk fermented with *Lactobacillus casei* subsp. casei R-68 contained 347.2 kJ of energy, 4.62% carbohydrates, 1.98% fat, and 2.08% protein, while soy-honey milk fermented with *Lactobacillus*
*plantarum* 1 R 1.3.2 contained 310.8 kJ of energy, 4.10% carbohydrates, 1.07% fat, and 2.60% protein per 100 g [28].

The studied soymilk products and fermented soymilk-honey were produced by the Agricultural Products Processing Laboratory of Riau University’s Agricultural College. After the soymilk sterilization at 115 °C for 10 min, it was immediately cooled to 45 °C. Then, 10% honey was added and inoculated with starter *Lactobacillus casei* subsp. casei R-68 (5%) and *Lactobacillus plantarum* 1 R. 1.3.2 (5%) in different soymilk media. Subsequently, the mixture was incubated at 37 °C for 18 h and products were ready to use.

### 2.3. Participants

This study was conducted in Pekanbaru City, Riau Province, Indonesia, from January to April 2021. On 19 October 2020, The Health Research Ethics Committee, Faculty of Public Health, Airlangga University, Surabaya, Indonesia, approved the study with registered number 98/EA/KEPK/2020. Subsequently, all participants had given written consent to participate before the study began.

Initial participants were 70, where 13 were excluded due to not meeting criteria of the study. An intervention period of 90 days was set due to changes in osteocalcin levels in response to interventions that could be monitored after three months [21]. Enumerators went from house to house, asking individuals for participation in the study. Subsequently, time after menopause, history of chronic disease, drug consumption, and soy allergy were all determined through interviews with participants. Inclusion criteria included the following: experiencing of menopause >4 years, preferring the taste of soymilk or fermented soymilk-honey, healthy, no history of intestinal diseases, bone, liver, kidney, hormonal disorders, or cancer, willingness to participate in the study for 90 days, and blood chemistry examination within normal limits (random blood glucose and uric acid). Meanwhile, exclusion criteria comprised of the following: taking osteoporosis drugs, hormone replacement therapy, corticosteroids, calcium supplements, and soy allergies. The participants’ cholesterol levels were controlled by checking initial and final cholesterol levels.

During the intervention period, participants were asked to avoid foods or probiotics supplements. Before the study commenced, the informed consent was discussed with the participants, and once they agreed to participate in the study for 90 days, the consent form was signed.

Eligible participants were required to visit the comprehensive elderly care service station twice (i.e., before and after the intervention) for several checks. Anthropometric measurement (weight and height), the 24-h food recall interview, and blood biochemical examination (random blood glucose, uric acid, total cholesterol, and serum osteocalcin level) were performed before and after the intervention. In addition, a 24-h food recall was performed in the middle of the study (day 45) by visiting the participants’ residences. To calculate the intake of calcium, phosphorus, and magnesium, the Nutrisurvey program was employed.

### 2.4. Intervention

The 57 participants were divided into three groups namely, the soymilk group (SM) consisting of 20 participants as a control, the soy-honey fermented milk group with *Lactobacillus casei* subsp. casei R-68 (SMH Lc) with 19 participants, and the soy-honey fermented milk group with *Lactobacillus plantarum* 1 R 1.3.2 (SMH Lp) with 18 participants. Participants received 100 ml of soy milk (SM) or fermented soymilk (SMH Lc or SMH Lp), and the enumerator questioned their compliance levels daily. In addition, all participants’complaints due to the intervention were monitored and recorded daily.

Participants received food packages (noodle, sugar, vegetable oil, and egg) and free blood checks at the beginning and end of the study to improve compliance during the intervention period.

### 2.5. Blood Sample Collection and Laboratory Analysis

The blood samples were collected and analyzed by staff from the Prodia Laboratory. Random blood glucose, uric acid, total cholesterol, and serum osteocalcin level were examined at beginning and end of intervention.

The random blood glucose levels were tested using the Hexokinase Method, while uric acid and cholesterol total levels were examined by the Uricase Method and Enzyme Colorimetric Method, respectively. Furthermore, the NMID test of osteocalcin used the electrochemiluminescence immunoassay (ECLIA) method. We used the N-MID level of osteocalcin because the concentration was more stable and accurate [18].

The primary outcomes were the change in the mean concentration of osteocalcin before and after the intervention for 90 days in the SM, SMH Lc, SMH Lp groups. Meanwhile, the secondary outcome was the difference in the mean blood serum osteocalcin levels after the intervention in the three groups. Other findings were also analysed such as changes in the mean of random blood glucose, uric acid, and total cholesterol levels before and after the intervention for 90 days in each group.

### 2.6. Statistical Analysis

Descriptive statistics were used to analyse the characteristics of participants. For continuous data presentation, the mean and standard deviation (SD) were used, while categorical data used frequency and numbers (%, n). The Kolmogorov–Smirnov test was used to examine data normality, while the t-dependent test was used to compare serum osteocalcin pre–post intervention. Furthermore, the one-way ANOVA test was used to compare the effect of the intervention on serum osteocalcin in postmenopausal women from each group. Based on the average decrease in serum osteocalcin, the intervention effect was observed, while the significance level (α) of 0.05 or *p* < 0.05 was used to examine a statistically significant difference or effect. Differences before and after the intervention for other outcomes such as random blood glucose, uric acid, and cholesterol were also examined by dependent *t*-test. The data were analyzed using statistical software, SPSS version 19.

## 3. Results

### 3.1. Baseline Characteristics

In the screening test, 70 participants volunteered to participate in this study. However, after conducting blood biochemical examinations and interviews, only 57 participants met the criteria. Moreover, at the end of the intervention, three participants dropped out because they did not participate in the final blood biochemical examination for reasons such as working out of town (*n* = 1) and returning to their hometown (*n* = 2). Therefore, 54 participants followed the study to completion (Figure 1).

During the 90-day intervention period, the compliance level of the participants was calculated from the returned milk bottles. The compliance was >80%, which was 88.05%, 88.76%, and 88.39% for SM, SMH Lc, and SMH Lp groups, respectively.

Based on the data normality test using the Kolmogorov–Smirnov test, data on age, weight, height, and calcium intake were not normally distributed (*p* < 0.05), while data on BMI, duration of menopause, magnesium, and phosporus intake were normally distributed (*p* > 0.05).

All baseline data had a normal distribution based on the Kolmogorov–Smirnov test, except the cholesterol baseline data had a normal distribution based on the skewness ratio between −2 to +2. Uric acid and total cholesterol data after intervention were normally distributed except random blood glucose levels. Data transformation was carried out first for data not normally distributed. The results of the transformation based on the Kolmogorov–Smirnov test on random blood glucose levels after the intervention showed a normal distribution. Therefore, it could be continued with the dependent *t*-test. Data on osteocalcin levels before and after the intervention had a normal distribution based on the Kolmogorov–Smirnov test.

The average age of the participants was 59 years, and the highest average age was in the SM group. Additionally, the participants’ education level was still low (elementary and junior high school) (62.9%) and most of the participants were housewives. The average nutritional status of participants was overweight (>25), while the SM group experienced the longest time after menopause with an average of 11 years (Table 1).

The results of a 24-h food recall analysis at the beginning, in the middle, and at the end of the study revealed that the average intake of calcium, phosphorus, and magnesium of the participants was far below the Nutrient Adequacy Rate (RDA) 2019 for calcium 1200 mg, phosphorus 700 mg, and magnesium 340 mg (Table 1). Furthermore, the calcium intake of the participants in this study was the same as that of postmenopausal women in Malaysia, the Philippines, and India, which was less than 500 mg/day [2].

### 3.2. Baseline Data of Blood Biochemical Examination

Baseline data on random blood glucose, uric acid, cholesterol, and osteocalcin levels in the three groups were not significantly different (Table 2).

### 3.3. Measurement of Blood Glucose Levels, Uric Acid, and Total Cholesterol, before and after Intervention

#### 3.3.1. Soymilk (SM)

There were no differences in the mean blood glucose, uric acid, and cholesterol levels before and after the intervention in the SM group. Meanwhile, random blood glucose levels increased after the intervention period but were not significant (Table 3).

#### 3.3.2. Soy-Honey Fermented Milk with *Lactobacillus casei* subsp. casei R-68 (SMH Lc)

The mean random blood glucose and uric acid levels at baseline and end of the intervention in the SMH Lc group were not significantly different, but cholesterol levels before and after the intervention showed a significant difference (*p*-value < 0.05). Uric acid and cholesterol levels decreased, while random blood glucose levels slightly increased but were not significant (Table 3).

#### 3.3.3. Soy-Honey Fermented Milk with *Lactobacillus plantarum* 1 R 1.3.2 (SMH Lp)

In the SMH Lp group, the mean random blood glucose, uric acid, and cholesterol levels before and after the intervention were not significantly different. The uric acid and cholesterol levels were stable, while blood glucose levels increased, but were not significant (Table 3).

### 3.4. Differences in Osteocalcin Levels at Beginning and End of the Intervention in the Three Groups

Blood serum osteocalcin levels in all groups presented a decrease after 90 days of the intervention period. However, a significant decline occurred only in the SMH Lp group (*p* < 0.05) (Table 4).

### 3.5. Differences in Osteocalcin Levels in the Three Groups after Intervention

The SMH Lc group experienced the lowest reduction in blood serum osteocalcin levels. However, based on the ANOVA test, osteocalcin levels in the three groups at the end of the intervention were not significantly different (Table 5).

### 3.6. Safety

All subjects could consume soymilk (SM) or fermented soymilk-honey (SMH Lc or SMH Lp), even though the taste was a bit sour. At the beginning of the intervention, two respondents in the SMH Lp group became unwell after consuming fermented milk but were still willing to participate in the study. However, as the days passed, there were no more complaints from both participants. During the 90-day intervention period, no fatal adverse events were reported from subjects in either the SM, SMH Lc, or SMH Lp groups.

## 4. Discussion

According to this study, fermented soymilk-honey with *Lactobacillus plantarum* 1 R 1.3.2 (SMH Lp) could significantly decrease osteocalcin levels, in addition to the fermented soymilk-honey with *Lactobacillus casei* subsp. *casei* R-68 (SMH Lc) could lower cholesterol levels. In contrast, no significant differences in random blood glucose, uric acid, cholesterol, and serum osteocalcin levels were observed before and after the intervention in the soymilk (SM) group.

*Lactobacillus casei* subsp. *casei* R-68 are bacteria isolated from curd (*dadih*), buffalo milk fermented in bamboo tubes, widely found in West Sumatra and Kampar Regency, Riau Province, Indonesia. This bacterium can lower cholesterol levels and prevent bowel cancer by binding to cancer-causing mutagenic substances [29]. Meanwhile, *Lactobacillus*
*plantarum* 1 R. 1.3.2 are lactic acid bacteria isolated from spontaneously fermented soymilk. These bacteria are short rods and are homofermentative, and are able to bind bile acids to reduce blood cholesterol levels [30]. In a preliminary study, fermented soymilk using *Lactobacillus plantarum* 1 R. 1.3.2 with 10% honey could increase the lactic acid bacteria (11.34 log/mL) to a higher degree than those using *Lactobacillus casei* subsp. *casei* R-68, (11.24 log/mL) [28]. In general, most fermented foods contain at least 10^6^ microbial cells per gram, with concentrations varying depending on origin and time of day the product was analyzed or consumed [31].

Previously, several studies have demonstrated the effect of adding honey to soymilk. The combination of honey, soy milk, and aloe vera increases the amount of *Lactobacillus acidophilus*. This beverage enhanced the nutritional value and taste [15]. The addition of 4% honey to Rayeb milk containing 50% cow’s milk and 50% soy milk increased *Lactobacillus acidophilus* counts [16]. Slacanac et al. reported that the addition of 5% honey to soymilk resulted in substantial growth of *Bifidobacterium longum* Bb-46. Furthermore, Stijepic et al. concluded that the addition of 4% honey to soymilk increased the viscosity, water-binding capacity but reduced the syneresis of soy yogurt. Honey is a functional food with unique ingredients, properties against pathogenic bacteria, and bifidogenic effects. It contains fructooligosaccharides, which act as prebiotics and has various nutrients [14,32]. The preliminary study supported earlier studies that fermented soymilk with 5% honey increased the growth of *Lactobacillus casei* subsp. *casei* R-68 and *Lactobacillus plantarum* 1 R 1.3.2. However, the panelists’ preference level increased with the addition of honey [28]. Based on this, 10% honey concentration was used in this study.

Animal and human studies have both reported the role of probiotics and prebiotics on bone health. Research in mice has shown that fermented soy skim milk reduced bone loss and the risk of osteoporosis [23]. Similarly, synbiotics (the combination specific to prebiotics with probiotics) stimulated bone mineralization and decreased bone turnover in rats. The effect of synbiotics on bone occurs through the prebiotic fermentation mechanisms in the large intestine and modulation of intestinal bacteria. Hence calcium absorption can be increased by improving the absorption surface [25]. Mice fed a synbiotic diet had greater cortical bone wall thickness than those fed a standard diet as a control. This could be because the synbiotics diet increased bone strength in mice [24]. Furthermore, meta-analysis studies show that probiotic supplementation could enhance lumbar bone mass in postmenopausal women [3], and a combination of three *Lactobacillus* strains prevented spinal bone loss in healthy postmenopausal women [26]. Also, *Bacillus subtilis* C-3102 increased bone mass in healthy postmenopausal women by reducing bone resorption and modifying gut bacteria [27]. Kefir-fermented milk consumption for more than six months increased hip bone mass in osteoporotic patients [4]. These results were similar to those in the current study.

The following are three possible mechanisms that explain the effect of fermented soymilk-honey with *Lactobacillus plantarum* 1 R 1.3.2 (SMH Lp) on osteocalcin levels. Firstly, during soybean milk fermentation, *Lactobacillus plantarum* 1 R. 1.3.2 and β-glucosidase convert isoflavone glucosides (daidzin, genistin, glycitin) into aglycones (daidzein, genistein, and glycitein). Isoflavone glucosides are poorly absorbed in the small intestine compared to aglycones because they have a larger molecular weight. Daidzein production enables intestinal bacteria to form equol and *O*-Desmethylangolensin (O-DMA). Equol is more estrogenic than O-DMA for bone metabolism in menopausal osteoporosis. Additionally, equol, which is primarily present as a glucuronide conjugate and binds to the estrogen receptor, leads to reduced bone resorption, enhanced bone formation, and increased bone biochemical and microstructural characteristics in individuals with menopausal osteoporosis. However, it does not influence healthy women that just began menopause [13,33,34]. Secondly, the oligosaccharides in soybeans and honey promote lactic acid bacteria fermentation and the production of large amounts of short-chain fatty acids, thereby lowering the colon pH. Thish as the potential to increase calcium absorption and, ultimately, bone mineralization. Furthermore, increased calcium absorption lowers parathyroid hormone (PTH), which in turn reduces calcium release from bones or inhibits bone loss [5]. Short-chain fatty acids are produced during fermentation by intestinal bacteria, and these fatty acids regulate bone formation and resorption [26]. Thirdly, gut bacteria regulate calcium absorption through intestinal pH changes and increase calcium solubility, hence supporting bone health [35].

Only the SMH Lp group experienced a significant drop in blood serum osteocalcin levels. Moreover, the osteocalcin levels in the three groups remained non-significantly different after the intervention and may be due to variations in equol production by gut bacteria. This production from dietary phytoestrogens varies significantly in individuals mainly due to intestinal microbial composition, notably in elderly subjects with low intestinal colonization and the possible correlation among the three kinds of phytoestrogens and dietary components [11,34]. Each individual’s ability to produce equol differs depending on their gut bacteria capacity to create equol. In Asian countries where the isoflavone daidzein is frequently consumed, only 40–60% of people produce equol [36,37]. The administration of soymilk or fermented soymilk increases the isoflavone metabolites (O-DMA and equol) in urine excretion [38]. In addition, in a complex intestinal microecology, different probiotics and prebiotics have different responses on distinct gut microorganisms that cannot be returned in the short term [39]. The influence of probiotics depends on the strain, dose, and ingredients that make probiotic products [40]. In this study, different probiotic administrations (*Lactobacillus plantarum* 1 R 1.3.2 or *Lactobacillus casei* subsp. *casei* R-68) may have various effects. Although not significantly, the SM and SMH Lc groups indicated a decrease in osteocalcin levels. However, by increasing the sample size and study duration, a significant intervention effect could be observed.

Another finding from this study is that fermented soymilk-honey with *Lactobacillus casei* subsp. *casei* R-68 can significantly lower total cholesterol levels. Pato et al. discovered that *Lactobacillus casei* subsp. *casei* R-68 and some lactic acid bacteria (LAB) from curd lowered cholesterol by a taurocholic acid deconjugation mechanism [29]. The results of this study were consistent with those of Feizollahzadeh et al., which explained that probiotic soymilk had a significant effect on the lipid profile of patients with type 2 diabetes mellitus, particularly reducing LDL cholesterol levels while increasing HDL cholesterol levels [41]. In addition, probiotics bring about metabolic effects through deconjugated and secreted bile salt, thereby decreasing serum cholesterol [40]. Significant reductions in cholesterol levels did not occur in the SM and SMH Lp groups. This could be due to differences in absorption and metabolism of isoflavones mainly related to characteristics in gut microbes [42]. Higher doses of soymilk or *Lactobacillus plantarum* 1 R 1.3.2 may be required to increase the effectiveness of the intervention in the SM or SMH Lp groups.

Random blood glucose levels increased in the three groups but were not significantly different. Sugar levels may need to be reduced in soymilk production to prevent a rise in blood glucose. Before and after the intervention, uric acid levels decreased, although they were not significant in the three groups. This suggests that the consumption of soymilk or fermented soymilk-honey does not increase uric acid production. Serum uric acid levels are increased in women undergoing both natural and surgical menopause. In the Chinese study on postmenopausal women with prehypertension or prediabetes, soy consumption did not affect uric acid levels. In addition, flavonoids such as genistein in soybeans inhibited the formation of uric acid by xanthine oxidase. However, other experimental studies in mice showed that serum uric acid levels were increased once administered with the flavonoids. Animal studies on isoflavones gave conflicting results, while in human studies, there was no effect on uric acid levels [43].

The health benefits of foods containing lactic acid bacteria have been recognized globally. Furthermore, lactic acid bacteria fermentation can either speed up digestion or improve the nutrients’ bioavailability of food. Different types of lactic acid bacteria and LAB fermented food products have evolved and provided benefits for humans and animals. These benefits include increasing the nutritional value of food, improving the ability to prevent infection, relieving symptoms of lactose intolerance, modulating the immune response, controlling cholesterol levels, and decreasing the risk of colon disease [44]. Moreover, fermentation improves the sensory characteristics of soymilk by reducing the unpleasant taste and improving the soymilk aroma and taste [12]. In this study, the fermented soy-honey milk is sour in taste and brownish due to the honey color. Hence, older participants preferred soymilk (SM) to fermented soymilk. After the 90-day intervention, participants gave various responses on soymilk with honey or fermented soy-honeymilk, for instance, the refreshing effect on the body (SM, SMH Lc, and SMH Lp group). Furthermore, increased appetite was observed in SM, SMH Lc, and SMH Lp groups, while pain in the joints was reduced in SM and SMH Lp. The SMH Lp group rarely experienced dust allergy, and reduced hip pain was observed for the SM group, while smoothed bowel movements were observed in SMH Lc and SMH Lp groups. Also, reduced headaches and chest pain was observed in the SMH Lc group, while the SMH Lp group experienced reduced finger numbness. Moreover, the administration caused good sleep (SM group and SMH Lc group), weight gain (SMH Lc group and SMH Lp group), rarely thrush (SMH Lp group), weight loss (SMH Lp group), and sleepiness (SM group, SMH Lp group). Correspondingly, the sour taste fluctuated in the SMH Lc group and some participants suggested that menopausal women continue to receive fermented soy-honey milk while reducing the acid taste. Similarly, dyeing of the fermented soy-honeymilk is required to improve its acceptance, hence it would not be tiresome.

The results of this study indicated that the biological availability of isoflavones and calcium absorption was increased by fermented soy-honey milk. Interestingly, soy-honey milk fermented with *Lactobacillus plantarum* 1 R. 1.3.2 reduced osteocalcin levels (*n* = 18), while soy-honey milk fermented with *Lactobacillus casei* subsp. *casei* R-68 lowered cholesterol levels (*n* = 17). Participants were recruited from house-to-house by enumerators in Tangkerang Labuai Village and were considered representative of the population. However, further research is required to confirm the effectiveness of the treatment.

The study’s uniqueness exists in the fact that soymilk-honey fermented with *Lactobacillus casei* subsp. casei R-68 and *Lactobacillus plantarum* 1 R 1.3.2 could potentially enhance health, particularly bone health and cholesterol levels. Moreover, honey contains various nutrients and bioactive components such as flavonoids, phenolic acids, enzymes, and many others that can be added to increase the nutritional value of honey. Soybeans and honey contain oligosaccharides, which act as prebiotics increasing the growth of *Lactobacillus casei* subsp. *casei* R-68 and *Lactobacillus plantarum* 1 R 1.3.2. The bioactive substances derived from the two products (soybean and honey) combined in one fermented milk product could have a synergistic effect that mutually strengthens their potential in maintaining health, particularly the bone tissue. However, there are several limitations in this study that need to be considered. First, this study used only one bone turnover marker, so it could not describe the bone health of menopausal women. Second, a non-randomized quasi-experimental design was used in this study. Therefore, random bias factors could occur and affect the study’s results. Third, the measurement of bone mass at the beginning of the study was not performed, hence the bone mass density status of the subjects was not identified (normal, osteopenia, or osteoporosis).

This study revealed that the treatment of soy-honey milk fermented with *Lactobacillus plantarum* 1 R 1.3.2 in postmenopausal women could significantly reduce the serum osteocalcin levels. However, soy-honey milk fermented with *Lactobacillus casei* subsp. *casei* R-68 could also decrease serum osteocalcin levels, but without significance. In conclusion, a more conclusive study is required with using more than one bone marker to determine bone health, a randomized controlled clinical trial design, increasing the sample size, extending the intervention time, and measuring bone mass density before the intervention.

## Figures and Tables

**Figure 1 nutrients-13-03581-f001:**
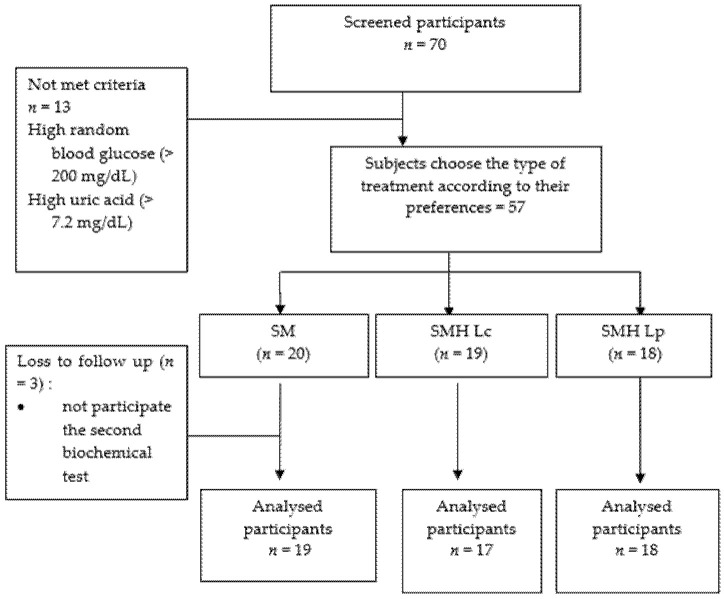
Flow diagram of subject recruitment and follow-up.

**Table 1 nutrients-13-03581-t001:** Participant Characteristics ^1^.

Characteristics	All	SM	SMH Lc	SMH Lp
	(*n* = 54)	(*n* = 19)	(*n* = 17)	(*n* = 18)
Age (year)	59.72 ± 7.68	65.16 ± 8.25	55.88 ± 4.53	57.61 ± 6.32
Education				
Elementary school (%)	48.1 (26)	63.2 (12)	47.1 (8)	33.3 (6)
Yunior High School (%)	14.8 (8)	10.5 (2)	11.8 (2)	22.2 (4)
Senior High School (%)	35.2 (19)	21.1 (4)	41.2 (7)	44.4 (8)
College (%)	1.9 (1)	5.3 (1)	0 (0)	0 (0)
Profession				
House Wife (%)	90.7 (49)	94.7 (18)	94.1 (16)	83.3 (15)
Employee (%)	1.9 (1)	5.3 (1)	0 (0)	0 (0)
Self-employee (%)	7.4 (4)	0 (0)	5.8(1)	16.7 (3)
Weight (kg)	59.44 ± 12.50	56.24 ± 14.31	61.65 ± 11.89	60.73 ± 10.91
Height (cm)	149.19 ± 5.95	148.47 ± 4.61	149.71 ± 8.12	149.44 ± 4.99
BMI (kg/m^2^)	26.69 ± 5.24	25.51 ± 6.19	27.44 ± 4.19	27.23 ± 5.09
Time after menopause (year)	9.09 ± 6.88	11.79 ± 9.07	6.53 ± 2.76	8.67 ± 6.18
Calcium intake (mg)	239.86 ± 169.35	232.68 ± 175.92	268.54 ± 199.28	220.33 ± 134.11
Phospor intake (mg)	586.88 ± 174	577.04 ± 179.88	605.87 ± 205.06	579.33 ± 141.51
Magnesium intake (mg)	177.34 ± 49.75	166.61 ± 39.46	182.89 ± 56.28	183.42 ± 53.73

^1^ Values in mean ± SD except for categorical data using frequency and numbers (%. *n*).

**Table 2 nutrients-13-03581-t002:** The baseline for Blood Biochemical Examination of the SM, SMH Lc, and SMH Lp groups.

Variable	*n*	Mean	SD	*p*-Value *
Random Blood Glucose (mg/dL)				
SM	19	104.95	19.72	0.68
SMH Lc	17	108.59	19.31	
SMH Lp	18	110.83	21.97	
Uric acid(mg/dL)				
SM	19	4.87	1.04	0.76
SMH Lc	17	4.82	1.13	
SMH Lp	18	5.07	1.00	
Cholesterol (mg/dL)				
SM	19	228.53	51.52	0.28
SMH Lc	17	236.82	34.72	
SMH Lp	18	214.78	33.23	
Osteocalcin (ng/mL)				
SM	19	31.60	12.47	0.55
SMH Lc	17	29.71	9.26	
SMH Lp	18	33.76	10.46	

* The *p*-value obtained using the ANOVA test.

**Table 3 nutrients-13-03581-t003:** Differences in Mean Levels of Blood Glucose, Uric Acid, Cholesterol Pre–Post. Intervention in the Three Groups.

Variable	*n*	Mean	SD	*p*-Value
SM				
Random Blood Glucose (mg/dL)				
Pre-test	19	104.95	19.72	0.28
Post-test	19	110.58	19.13	
Uric Acid (mg/dL)				
Pre-test	19	4.87	1.04	0.57
Post-test	19	4.78	0.99	
Cholesterol (mg/dL)				
Pre-test	19	228.53	51.52	0.52
Post-test	19	224.21	48.55	
SMH Lc				
Random Blood Glucose (mg/dL)				
Pre-test	17	108.59	19.31	0.75
Post-test	17	110.12	24.66	
Uric Acid (mg/dL)				
Pre-test	17	4.81	1.13	0.75
Post-test	17	4.77	1.21	
Cholesterol (mg/dL)				
Pre-test	17	236.82	34.72	0.02 *
Post-test	17	222.24	34.42	
SMH Lp				
Random Blood Glucose (mg/dL)				
Pre-test	18	110.83	21.97	0.12
Post-test	18	122.94	32.54	
Uric Acid (mg/dL)				
Pre-test	18	5.07	1.01	0.69
Post-test	18	5.01	1.18	
Cholesterol (mg/dL)				
Pre-test	18	214.78	33.23	0.69
Post-test	18	215.89	34.36	

* The difference before and after the intervention was significant *p* < 0.05 (dependent *t*-test).

**Table 4 nutrients-13-03581-t004:** Differences in Mean Osteocalcin Levels (ng/mL) at Beginning and End of the Intervention in the Three Groups.

Variable	*n*	Mean	SD	*p*-Value
SM				
Osteocalcin				
Pre-test	19	31.59	12.46	0.40
Post-test	19	30.64	11.44	
SMH Lc				
Osteocalcin				
Pre-test	17	29.71	9.26	0.21
Post-test	17	27.74	8.81	
SMH Lp				
Osteocalcin				
Pre-test	18	33.76	10.46	0.02 *
Post-test	18	29.34	12.54	

* The difference before and after the intervention was significant. *p* < 0.05 (dependent *t*-test).

**Table 5 nutrients-13-03581-t005:** Differences in Mean Levels of Blood Serum Osteocalcin (ng/mL) after the Intervention in the Three Groups.

Variable	*n*	Mean	SD	*p*-Value *
SM	19	30.64	11.44	0.73
SMH Lc	17	27.74	8.81	
SMH Lp	18	29.34	12.53	

* The *p*-value obtained using the ANOVA test.

## Data Availability

All relevant data analyzed during the current trial are included in the article. Access to raw datasets may be provided upon reasonable request to the corresponding author.

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
