# Peer review of "Effect of Fermented Soymilk-Honey from Different Probiotics on Osteocalcin Level in Menopausal Women"

_nutrients, 2021, doi:10.3390/nu13103581_

Round 1
Reviewer 1 Report
The aim of this paper is interesting, however, the study is not well planned, especially in terms of bone turnover assessment. To evaluate bone metabolism, not only bone formation markers (e.g. osteocalcin) but also bone resorption markers (e.g. cross linking telopeptide of type I collagen – CTX, NTX) should be measured. If bone formation and resorption markers had been determined, it would possible to analyze the ratio of bone formation to bone resorption processes (i.e. OC/CTX). Additionally, determination of osteocalcin forms such as carboxylated osteocalcin and undercarboxylated osteocalcin (which play different role in bone or glucose metabolism) would be valuable. Besides, the study was conducted for a short period of time (90 days).
I have the following comments and questions with regard to specific sections of the manuscript:
Abstract:
- Please change abstract to more clear and informative.
- The conclusion in this section should be modified (it is not exactly based on the results of this research).
Introduction:
- In the aim of this study it says: “For this reason, this study aimed to examine the effect of fermented soy-honey milk using different probiotics (Lactobacillus 79 casei subsp. casei R-68 and Lactobacillus plantarum 1 R.1.3.2) on osteocalcin levels in postmenopausal women. This protein is produced by osteoblasts and is used as a marker to evaluate bone turnover”.
To evaluate bone turnover, it is not enough to measure osteocalcin, but to measure markers of bone resorption.
Methods:
- Please clarify: In Methods: 57 participants were divided into three groups namely, the soymilk group (SM) consisting of 20 participants as a control, the soy-honey fermented milk group with Lactobacillus casei subsp. casei R-68 (SMH Lc) with 19 participants, and the soy-honey fermented milk group with Lactobacillus plantarum 1 R 1.3.2 (SMH Lp) with 18 participants.
In Results: 54 participants.
- What statistical software was used to analyze obtained results?
- A multivariate regression analysis should be performed with a detailed description of the statistical models used.
Results:
- This section should be changed.
- All data were presented as mean values and SD. Did all data have normal distribution?
- Groups are not homogeneous, e.g. SM group: oldest, longest menopause, lower BMI.
- When differences are statistically insignificant please do not indicate p>0.05
- Table 1: Please correct: “Phospor intake”, description under table 1.
- Results in the Table 5 is repetition those in Table 4
Discussion:
- Limitation and conclusion should be changed.
References:
- References should be written uniformly and in accordance to Instructions for Authors.
- The authors sometimes write the whole title of the journal and sometimes just the abbreviation. Some words in article titles are capitalized and some are lower case.
- I think that DOI should be added.
Reviewer 2 Report
The manuscript entitled, "Improving Bone Health in Menopausal Women with Fermented Soymilk-Honey from Different Probiotics" provides a direction of possible use of synbiotics made of soymilk, honey and bacteria for improving the bone health in menopause women. However, for further improvement of manuscript additional evidences are required.
Abstract:
1. Would be better to mention “blood serum osteocalcin levels” instead of only “osteocalcin levels” in line 20.
Introduction:
1. Would be better to add some information on effects of soy milk and honey in Lactobacillus genus bacteria either in Introduction or discussion part.
Materials and Methods:
1. It would be better if the author could have made this section well organized. It would be better to start with study design, preparation of synbiotics, participants, interventions, blood sample collection and laboratory analysis, and statistical analysis. Outcomes could be mentioned in laboratory analysis and sample size in the study design.
2. In the flow chart, initial participants are 70 so it would be better to mention at line 91 that the study included 70 participants, where 13 were excluded due to not meeting the criteria of the study.
3. The authors have mentioned that the participants' cholesterol levels were controlled before the intervention started and after the intervention. It would be better if the authors could inform how the levels were controlled.
4. It would be better if the authors could mention the honey used was raw or purified. Additionally, the authors could provide information how the bacteria were obtained.
5. In the line 149, the authors have mentioned that the participants received food packages. It should be clearly mentioned about the foods included. Was the food recall analysis done from that food packages?
6. The authors have examined the calcium and phosphor intake through the food recall analysis. It would be interesting to see the blood serum calcium and phosphate levels before and after intervention.
7. Menopause women have higher risks of bone fracture. Was there any bone fracture incidence in any of the participant?
Discussion:
1. Would be better to discuss menopause, osteocalcin level and osteoporosis correlation.
2. The authors have mentioned most of the mechanism of action as possible mechanisms but could not show any evidence for at least one or two. The authors themselves have agreed the limitations of the study that it does not have bone mass and density results of the participants. However, only with osteocalcin levels it is not sufficient to conclude the findings. It would be better if the authors could have examined serum levels of parathyroid hormone, bone turnover markers including N-terminal propeptide of type I procollagen (PINP), and C-telopeptide of type I collagen (CTX-I), calcium, phosphate levels, and etc to conclude the mechanism.
3. Osteoporosis in menopause women may coexist with other metabolic disorder diseases. The authors examined the random blood glucose level, uric acid and cholesterol levels of participants but have not discussed the association. It would be interesting if the authors could have discussed how these factors are correlated with bone health and osteoporosis in menopause women.
4. Most of the statements in the discussion seems to be unrelated to the main aim of the study. Would be better to include those findings (sleep, weight gain, weight loss, sleepiness, thrush, taste, and etc) in the result as levels or frequency or percentage of participants and discussed how they were affected by intervention and finally the bone health.
5. Lines 464-465 do not seem appropriate.
6. Citation for lines 476-480 is missing.
7. In the conclusion, the authors have suggested that Lactobacillus plantarum 1 R 1.3.2 could be used for fermentation of any product consumable to menopause women. Without any evidence of fermentation of other products and impact on health, it does not seem appropriate to make the final suggestion.
8. The discussion section gives a lot of information but it is too long and has loss of the focus of the study in many places. It would be better if the authors had discussed precisely and focusing on the main aim of the study.
Reviewer 3 Report
The title of submitted manuscript is missleading – one can expect that bone health was assessed in this study while in fact it was not. I have several comments in the relations to the design of the study.
Most of all- bone health cannot be assessed based only on the levels of one marker (osteocalcin) which in addition is not generally accepted as a reference bone formation marker for investigation in humans. Bone health may be assessed by bone mineral density and specific markers of bone turnover which are: P1NP for assessment of bone formation and CTX for bone resorption. These two are generally accepted as reference markers for monitoring changes in bone turnover as the age and sex reference values for them are established and CV for their biological variation as well. Calcium and phosphorus levels do not say much on bone health either.
Besides, P1NP and CTX can be assayed on the same analyzer as NMid osteocalcin therefore the question arises why the authors have chosen osteocalcin instead P1NP.
The methods used are very poorly described.
Why random glucose (non-fasting) instead of fasting glucose was measured. This rises another question whether totsal cholesterol was also measured in the non-fasting state ? From the clinical point of view the data on circulating biochemical parameters seem to be doubtful.
Round 2
Reviewer 2 Report
Dear authors,
Appreciate efforts made by authors to improve the manuscript. Now, the title seems appropriate to the manuscripts according to the findings made. Still there are minor points to be addressed.
- As mentioned earlier, "cholesterol levels control". The authors do not seem to apply any intervention for controlling the cholesterol levels in participants as they have just informed that it was checked(seems like examination) at initial and final. So it would be better to mention that cholesterol levels were analyzed and participants with normal or certain ranges were included in the study.
- In the previous version at lines 464-466 and present version lines 911-912 it is mentioned, "Few participants even gave suggested that fermented soy-honey milks should be produced for trading". It does not seem relevant.
Reviewer 3 Report
The title and the content of the manuscript have been changed according to the reviewers’ suggestions.
However, the statement on the osteocalcin to be used in monitoring of antiresorptive therapy is not acceptable as this is not true. The authors have to comment on that.
In 2017 Int Osteoporosis Foundation and the Eur Calcif Tissue Soc published recommendations on the bone markers to be used in monitoring antiresorptive therapy with oral bisphosphonates which are most widely used (Osteoporos Int 2017, 28: 767). The recommended markers are P1NP (bone formation) and CTX (bone resorption). Therefore the authors should mention that the choice of bone marker for monitoring therapy may depend on the type of antiresorptive agent used and in some situations osteocalcin can be used, however this is P1NP which is a reference marker for assessment of bone formation in adults and currently also in children and adolescents.
